# Study of Lapping and Polishing Performance on Lithium Niobate Single Crystals

**DOI:** 10.3390/ma14174968

**Published:** 2021-08-31

**Authors:** Karim Ravilevich Muratov, Timur Rizovich Ablyaz, Evgeny Anatolevich Gashev, Sarabjeet Singh Sidhu, Evgeny Sergeevich Shlykov

**Affiliations:** 1Mechanical Engineering Faculty, Perm National Research Polytechnic University, 614000 Perm, Russia; karimur_80@mail.ru (K.R.M.); kot_ostrow@mail.ru (E.A.G.); kruspert@mail.ru (E.S.S.); 2Mechanical Engineering Department, Sardar Beant Singh State University (Formerly, Beant College of Engineering and Technology), Gurdaspur 143521, India; sarabjeetsidhu@yahoo.com

**Keywords:** lithium niobate single crystal, abrasive lapping, polishing, surface roughness, deformed layer, tool wear, kinematics

## Abstract

Recently, the range of crystal materials used in industrial microelectronics has significantly increased. Lithium niobate single crystals are most often used in integrated optics, due to the high values of optical and electro-optical coefficients. An integral-optical circuit based on a lithium niobate single crystal is a key element in the production of local high-precision fiber-optic gyroscopic devices used in civil and military aviation and marine technologies. In the process of production of an integral-optical circuit, the most labor-intensive operations are mechanical processing, such as lapping and polishing. Technological problems that arise while performing these operations are due to the physical and mechanical properties of the material, as well as target surface finish. This work shows the possibility to achieve the required surface quality of lithium niobate single crystal plates by mechanization of lapping and polishing process in this article.

## 1. Introduction

Recently, the range of crystal materials used in industrial microelectronics has significantly increased. One such material is lithium niobate single crystals (LNSC). In integrated optics, due to their high optical and electro-optical coefficient values, LNSC are most often used.

The main processes for obtaining accurate and clean surfaces in the production of optical components are finishing and lapping with abrasive powders in a free or fixed state. The deformation mechanism during abrasive processing of various optical materials is not the same and depends on their physical and mechanical properties [1,2,3,4,5]. For instance, while processing plastic based optical materials, the underlined material removal phenomenon is the removal of material in the form of thin layers, thus the appearance of plastic deformation in neighboring areas. However, while processing crystalline materials like germanium silicon and glass, the resulting microcracks penetrate to a certain depth from the surface and form a deformed subsurface layer [6,7]. Thus, the abrasive processing of plastic and crystalline/brittle materials differ significantly.

Many studies have been devoted to the research of the phenomena underlying the processes of abrasive processing of brittle optical materials. That most of the studies of abrasive dispersion are mainly carried out on glass material. In comparison to glass, most crystalline substances are much lower in mechanical properties. However, few have high strength and hardness, such as optical ceramics. The theory of the impact of free abrasive grains on the glass surface during its lapping is now generally accepted and has found its further development and refinement [6,8,9,10,11,12,13]. There are some differences in the operation of free and fixed abrasive grains which significantly influence the mechanism of action, the size of the deformed layer, and the micro relief of the processed surface [9,14,15]. In studies [1,9,10,16], it was found that the ratio of the depth *F* of the deformed and h of the relief layers for various grades of optical glass and minerals with a similar brittle criterion remains almost constant: *F*/*h* = *K* ≈ 4 (Figure 1).

The size of the abrasive grain and its hardness have the most significant impact on the depth of the relief layer. In contrast, the hardness of the lapping material, pressure, and speed of finishing have a negligible impact. During the abrasive processing of crystalline materials, for example, single crystals of lithium fluoride and fluorite, the direction of cracks propagation and resulting strain differs from their direction in the glass and depends on their structure. Therefore, the microrelief of lapped surface of crystals is also related to the structure and the crystallographic direction in which the lapping is preformed [1,17].

The process of polishing brittle optical materials removes the deformed layer caused by lapping and surface texture whose defects are significantly less than the wavelength of light (about 1/4–1/8 λ) also creates. As a result, the surface becomes transparent and perfectly smooth. After polishing, the deformed layer of crystalline materials is also heterogeneous and has a complex structure [10,11]. Even after chemical-mechanical polishing using ultra and nano-disperse diamond powders [11], the nano scratches of various types were detected on the processed surface: dash-point or dot-shaped, short and long, straight and arc-shaped. Below the relief zone area is an elastic-stressed, where surface structural defects may occur. The sum of these areas determines the depth of the deformed layer resulting from polishing.

The abrasive finishing of end surfaces of LNSC plates is not found in the literature so far. Hence, the question of the depth of the deformed layer and the features of precision processing of modern single-crystal materials, such as LNSC, remains almost unstudied. Fine-turning and polishing small areas in relation to the lapping area causes local tool wear. This affects the quality of the processed surface integrity. The uniform distribution of processing traces depends on the size of the surface to be processed and the tool, and development of technology for mechanized lapping and polishing of optical materials to ensure good quality.

In the present study, an attempt has been made to improve the efficiency of lapping and polishing by mechanizing the process of abrasive finishing of the ends LNSC plates. To achieve the same, the effect of the trajectory of tool on the quality of surface is investigated and quantified by the depth of the deformed layer. Furthermore, the possibility of achieving the required surface quality of the LNSC plates based on the mechanization of the polishing process is analyzed and quantitative and qualitative indicators are established.

## 2. Material and Methods

LNSC plates after diamond cutting were used as samples in experiments. The length dimensions of the workpiece is 36 × 3.04 × 12.84 mm (l × w × h). Processing of the end faces of the plates was carried out in packages of 4–5 pieces in a special fixture on the finishing and polishing machine (Innovation 200R Twin, Remet, Italy). The necessary pressure on the tool–work surface contact was generated and changed by calibrated weights. The original raw and processed surfaces of the ends were evaluated visually and by optical microscopy (Olympus GX51 and CAM-MS-01. USB). Surface roughness parameters Ra, Rz, and Rmax were measured using a profilometer (Perthometer S2, Mahr GmbH, Germany) and profilograms were recorded to assess of the microrelief. Measurements of the geometric form deviation of the processed LNSC plates and the tool were performed using form tester (MarForm MMQ 400, Mahr GmbH, Germany). The productivity of the process (removal rate) was estimated by the value of linear removal of the material per time *Q* (µm/min). A vertical optical long-range meter IZV-2 (USSR) with a division of 1 µm (Figure 2) was used to measure the amount of removal. This allowed rapid and minimal measuring effort for high accuracy.

Indicator head 2 type 1MIG, GOST 9696-82 with a division of 1µm, is fixed to the case device 1. The device is based on the inner diameter A and the ends surface B of the case fixture 3.The tip 4 of the indicator 2 is oriented above the package of workpieces 5 and the measurement is carried out. The material’s removal rate (Δ*_Amid_*, µm) is defined as the difference between the indicator readings before and after processing. To assess of the depth of deformed layer of the plate ends, electron scanning microscopy was used (SEM Hitachi S-3400N, Japan). Before analyzing the samples in SEM, the surfaces of the same were coated with a layer of gold with a thickness of 1 nanometer. Figure 3 shows the installation scheme of sample (1) on table (2) in the working area of the microscope (1).This installation scheme provided the determination of the width and depth of the deformed layer. Setup scan parameters are an accelerating voltage-20 kV, zoom from ×350 to ×2500, secondary electron mode, mode of electron backscattering.

Suspensions were prepared by mixing micro-powders made of white electrocorundum with a grain size of 24A M14, 24A M7, 24A M3 in deionized water lapping with a free abrasive. The concentration of the abrasive suspension by weight was 1/5. The base suspension in comparative experiments was 24A M7.

Polishing was performed on suspensions based on deionized water and synthetic diamond micro-powders ASM with a grain size of 1/0.5 and 0.5/0. The concentration of the suspension by weight was kept 1/10. The study of the lapping process using suspensions was carried out on round lappers (Russia) with a diameter of 200 mm made of optical glass grades K8 and LK5. Diamond lapping ASN 10/7-B3-01-2 and ASN 10/7-M2-01-2 were used for lapping with a fixed abrasive. Woven and non-woven hydrophobic materials were used as polishers in experiments were raincoat fabric (Bologna); nylon mesh with a thickness of 0.09 mm, cell size ~0.1 mm; polyethylene wrap with a thickness of 0.14 mm; raincoat fabric (80% polyamide, 20% polyurethane); awning synthetic fabric with a water-repellent coating; VelTex Buehler polisher (Switzerland); MicrolapRemet polisher (Italy); synthetic leather SK-8.The durability of the lapping working surface was evaluated by the running time between edits. The durability of polishers was also estimated by the total operating time before the appearance of local physical wear on the work surface or before a significant decrease in the polishing ability and the formation of visible marks on the processing surface.

## 3. Results and Discussion

Firstly, the influence of the type of trajectory on the lapped surface quality and the process productivity was determined. Machines with a raster and cycloidal trajectory movement, as well as rotational tool movement, were examined. A multi-position fixture was used to hold the workpieces and allow them to move along the lapping surface. Three sets of plates were installed in the position of the fixture. The size of the processed plates was 36 × 3 × 1 mm. Two types of lapps were used in the experiments.

A glass disk made of optical glass of the K8 brand was used with a suspension of white 24A electrocorundum with a grain size of M7 µm in deionized water, with a concentration of 1/5. The lapp with the fixed abrasive ASN 10/7-B3-01-2 mark was used with water as a cutting fluid. The constant processing modes were: cutting speed = 0.18 m/s, pressure P = 64 kPa at a clamping force of 0.24 N. Processing was performed for 1 min. Table 1 shows the mean values of output processing parameters.

The processing performance was increased by 25–30% when the kinematics of the lapping motion becomes more complex with any finishing method, but the surface roughness does not change significantly. The dependence of the increasing debris during the trajectory was examined with optical microscopy of the plate ends before and after processing. The largest spalls/debris (with a depth of more than 45 µm) after processing with a fixed abrasive on the Raster 220 lapping machine was detected. As a result, machines with a tool working movement with alternating speeds were not recommended for lapping operations of such fragile materials as LNSC.

The satisfactory results on the quality of LNSC plates were obtained during lapping on a machine with a rotational lapp movement. No large spalls were detected, regardless of the type of tool used. According to technical requirements, the roughness of the processed surface should not exceed 0.003 µm in the Ra parameter, while individual scratches on the surface and edges are not allowed. The main requirements for the plates fixture and parameters set up in the lapping machine were formulated by taking into account the high brittleness of materials and a small area of the processed surface:(1)To increase the area of the processed surface, it is necessary to process the plates in a set of 3–5 pieces;(2)To self-positioning of the fixture with workpieces on the lapp surface, the number of sets must be at least three;(3)Information about the acceptable deformation of LNSC plates was necessary for the development of their mechanical fixing;(4)The use of standard pressure spring or pneumatic mechanisms of lapping machines to create working pressure was not allowed due to the high brittleness of the material;(5)Processing with the workpieces beyond the lapp was not allowed to avoid the appearance of spalls on the processed surface and edges;(6)Setting parameters must ensure minimum possible uniform wear of the lapp;(7)Local wear of lapp should not exceed 20 µm so as to reduce the probability of the edge spalling and processed surface damage by fragments of the material.

A full factorial experiment was completed based on the stated requirements for fixture and setting parameters. The regression equation was obtained and allows us to estimate the value of local lapp wear
*y* = 8.389 + 5.461·*x*_1_ + 0.54·*x*_2_ + 1.313·*x*_3_(1)

The values of factors are used for decoding
*x*_1_ = 2·(*r* − *r*_0_)/(*r_max_* − *r_min_*), *x*_2_ = 2·(*n* − *n*_0_)/(*n_max_* − *n_min_*), *x*_3_ = (2·(*r* − *r*_0_)/(*r_max_* − *r_min_*))^2^,(2)
where *r*—radius of the workpieces location in the fixture, mm (limits: *r_min_* = 25 mm, *r_max_* = 44.5 mm); *n*—frequency of lapprotation, rpm (limits: *n_min_* = 42 rpm, *n_max_* = 150 rpm ); *r*_0_ = (*r_max_* + *r_min_*)/2—the main factor level, mm; *n*_0_ = (*n_max_* + *n_min_*)/2—the main factor level, rpm.

Substituting (2) in (1) we get a mathematical dependence
∆*h* = 55 + 0.06·*r*^2^ − 3.61·*r* + 0.01·*n*,(3)
where ∆*h*—working tool surface wear, µm.

Equation (3) is reduced to a dimensionless form for ease of use
∆*h/h_alw_* = 1100 + 1450·(*r/r*_0_)^2^− 2509·*r/r_0_* + 19.2·*n/n*_0_,(4)
where *h_alw_* = 20—the value of allowable wear, µm.

Dependence of tool wear on the fixture radius and the rotation speed test to the Fisher criterion with a confidence probability of 95% satisfied. On the Figure 4 show the response surface graph depending on the influence of the changed process parameters on the output response.

The minimal local tool wear was achieved when the workpieces are located in the fixture at the radius of *r* = 30 mm. However, it increases by 13 times when this value was increased to the maximum value of 44.5 mm (in this case, the diameter of the fixture covers half the diameter of the lapp).

When the rotation speed increases by more than 3.5 times, the depth of local tool wear increases by 2–3%. The lapp rotation speed has a less significant effect on the local tool wear. Thus, to ensure high productivity, the lapp rotation speed (as well as the cutting speed) should be selected as high as possible.

The detecting algorithm of the processing traces distribution on the lapp surface was designed to determine the nature of tool wear. The point moves according to the specified motion law and intersects the conditional grid on the lapp surface (Figure 5). Each point hit in a specific cell was registered. Accumulation of information about the number of intersections occurred when hits were repeated. Processing traces distribution field on the tool surface is the result of algorithm execution. The wear nature can be determined based on the analysis of the processing traces distribution field (Figure 5).

The software was designed to solve the problem of the uniform distribution of processing traces on the lapp surface. It allows predicting tool wear depending on the setting parameters (the size of the fixture and lapp, the displacement of rotation axes), kinematic parameters (the trajectory type and speed of relative movement), and other parameters such as the material of the workpiece and tool, granularity, concentration, type of excipient, working pressure) of the lapping process. The software was working using the Python programming language on the PyQt5 platform using the Matplotlib library (Figure 6a–c). The results given by the software have been experimentally verified under real processing conditions. The deviation of the geometric tool form was as a result of real processing coincides with the modeled deviation both qualitatively (local wear in the center) and quantitatively (the wear value is 4.5 µm) (Figure 6c and Figure 7). The universal multi-position fixture was designed and manufactured based on the results of a factor experiment and modeling of various setting parameters in the software. The fixture allows you to provide mechanized lapping and polishing of the plates ends by sets and increase the production program of finished parts by 12 times. A sustainable of geometric form deviation within 1 µm was achieved using the fixture. This is three times more than manual processing.

Plate sets in the fixture were located at a radius of 30 mm and act as a correct ring, which contributes to more uniform lapp wear. The task of lapping study was to establish the process conditions that ensure a productive removal of the allowance and to obtain a working surface with minimal mechanical defects before the polishing operation. A grain size larger than M14 while maintaining the same concentration leads to the number of abrasive grains decrease in contact with the tool workpiece. It leads to an increase in local forces on individual grains and the appearance of scratches and spalls on the processed surface (Table 2). The removal productivity increases linearly by 2.5–3 times with increasing pressure (Figure 8a) and cutting speed (Figure 8b). At the same time, the roughness of the processed surface changes slightly.

Deep scratches, cracks and spalls on the processed surface occur when the pressure was exceeded due to the brittleness of the LNSC. In addition, the abrasive suspension was torn from the lapp surface as a result of the action of centrifugal force when the rotation speed increases by more than 120 rpm. It can lead to ‘dry’ friction and analogous defects.

Processing with a fixed abrasive has several advantages compared to lapping with abrasive suspensions. Processing with a fixed abrasive, the productivity and surface roughness were located between the results obtained after lapping with suspensions 24A M3 and 24A M7 (Table 3).

Analysis of the results showed that the removal productivity increases almost proportionally when the pressure increases three times. The removal productivity increases by 3.5–4 times when the cutting speed increases from 0.22 to 1.1 m/s. Changes in the pressure and cutting speed do not significantly affect the roughness. In addition, the organic excipient B3-01, which was mainly, contains rubber flour as filler and used for finishing lapping, increases the polishing effect and reduces roughness.

The straightness deviation of the plates ends after lapping on glass lapp of the LK5 mark with 24A suspensions with a grain size from 3 to 14 µm at a length of 12 mm did not exceed 0.3–0.5 µm. The straightness deviation of the plates ends for lapp ASN 10/7-M2-01-2 did not exceed 0.4–0.5 µm.

The depth of the deformed layer F of the plate’s ends of a LNSC was studied by using electron microscopy in the mode of electron backscattering. The SEM of the deformed layer of LNSC after 6 min of lapping on K8 mark glass with 24A suspension, M3, M7, and M14 µm is shown in Figure 9.

Studies showed that the deformed layer depth *F* of LNSC is constant during lapping with a specific grain of powder. It confirms the high stability of the lapping process of LNSC. Also, studies showed that the structural defects were located at a small angle to the surface. This was due to the predominant direction of lapp movement on the sample.

Figure 10 shows the dependence of the deformed layer depth *F* on the height of the relief layer *h*. Studies showed that this value correlates with the surface roughness parameter Rmax. Studies of the deformed layer depth Fafter LNSC plate ends lapping were carried out on lapp of optical glass of the K8 mark with a suspension of 24A—1/5 grain size M3, M7, and M14 under the basic processing modes: Vmid = 0.48 m/s; P = 31 kPa.

The obtained dependence allows us to determine the value of the ratio of the deformed layer depth *F* and the height of the relief layer *h*. This ratio is *F/h* = 12 in the crystallographic direction during lapping of the LNSC plates ends. This conclusion allows us to estimate the deformed layer value by rapid measurements of the processed surface roughness (Rmax) of LNSC using a portable profilometer in the production environment. This makes it possible to exclude studies on expensive time consuming equipment such as electron microscopy. The ratio of the deformed and relief layers during processing with a fixed abrasive was determined in the same way. During processing on the ASN 10/7-B3-01-2 lapp, the ratio is also 1:12. Deformed by lapping layer requires removal during the subsequent polishing operation of the single crystal. The formula was recommended to calculate the total deformed lapping layer depth (F): F = 12·Rmax, µm. The presented results was in line with the conclusions drawn by the authors [3,4,5,7,9].

The purpose of polishing was to remove the material that exceeds the size of the deformed layer after the lapping operation and to obtain a surface free from mechanical defects. The results of comparative experiments on polishing productivity and surface quality are shown in Table 4. The polishing process durability was estimated by the total machine operating time Σ*t*_mach_ before the appearance of local centers of physical wear on the work surface or before a significant decrease in the polishing ability and the appearance of visible scratches on the polished surface.

Table 4 shows that synthetic woven hydrophobic materials such as raincoat fabrics, and non-woven polishers Microlap and VelTex, showed the highest removal productivity. However, woven material polishers are slightly worse than non-woven ones in roughness parameters. The higher elasticity and uniformity of the structure of non-woven materials provides a more uniform grain appearance in the abrasive layer, and good ‘adaptability’ of polishers to the processing surface.

Comparing the total machine operating time Σ*t*_mach_ (Table 4) shows that during the processing of plates ends surface with a small contact area, the durability of the polisher is determined not only by the strength of the material but also by its structure. As a result, of the impact of sharp edges of the processed plates, the fibers of polishers are deformed; local breaks and cuts appear on the surface of the polisher. Non-woven polishers with a fleecy structure (synthetic leather SK-8, MicrolabRemet and VelTex Buehler polishers) showed the highest durability with a diamond suspension grain size of 1/0.5 and a concentration of 1/10 in the basic modes. During polishing on thicker and softer (fleecy) polishers (raincoat fabric, Microlap, or VelTex), reducing the suspension granularity to 0.5/0 and significantly reduces the speed of polishing with a simultaneous decline in surface quality (defects in the form of holes and swells).

The polishing productivity was almost directly proportional to the average cutting speed. A four-fold increase in speed has almost no effect on the polished surface roughness. The speed limit was the same as for lapping—the risk of ‘dry’ friction and the durability of the polisher.

The purpose of the polishing operation was to create the required roughness and remove the layer deformed by lapping. Figure 11 shows that the required roughness is set after 2–3 min of processing. However, the deformed layer will not be completely removed during this time. The minimum polishing time was calculated based on the removal rate during polishing, the maximum depth h of the relief layer (Rmax), and the ratio of the depth of the deformed F for a LNSC was 1:12.

So, during the abrasive finish processing of crystalline materials (plate’s end of LNSC) the total depth of the deformed layer by lapping and the minimum polishing time required for its removal was calculated. The formula recommended is *F* = 12·Rmax, µm; *T_pol_* ≥ *F*/*Q_pol_*, min.

In the production environment, the calculation of the minimum polishing cycle time allowed to increase the processing efficiency by reducing the time of the polishing operation, reducing the number of polishing defects, and reducing the abrasive materials consumption. The presented results coincided with the authors’ conclusions [3,4,5,11,12,13,14].

## 4. Conclusions


The tool movement kinematics have shown that for lapping and polishing of brittle materials such as lithium niobate single crystal, it is advisable to use machines with a rotational movement. Machines with oscillating or translational tool movement are not recommended for plates ends finishing operations due to the high rate of change in the speed and acceleration vector direction.The minimal local tool wear is achieved when the workpieces are located in the fixture at the radius of *r* = 30 mm; however, increases by 13 times when this value is increased to the value of 44.5 mm. When the rotation speednincreases by more than 3.5 times, the depth of local tool wear increases by 2–3%. Therefore, to ensure high productivity, the lapp rotation speed (as well as the cutting speed) should be selected as much as possible.The designed software allows to prediction tool wear depending on the setting parameters (the size of the fixture and lapp, the displacement of rotation axes), kinematic parameters (the trajectory type and speed of relative movement), and technological parameters (the material of the workpiece and tool, granularity, concentration, type of excipient, working pressure) of the lapping process.The plates lapping with free abrasive results have shown that the removal productivity increases linearly with increasing pressure and cutting speed. The roughness of the processed surface changes slightly. Acceptable pressure values depend on the type and grain size of the abrasive and the suspension concentration. While using a 24A M14 suspension of one-fifth concentration, the permissible pressure is not more than 40 kPa. While using 24A M7 and 24A M3 suspensions of the same concentration, the pressure can be increased to 80 kPa. Increasing the abrasive grain size larger than M14 µm is impractical. The recommended lapp rotation speed should not exceed 120 rpm.For processing brittle materials, such as lithium niobate single crystal, it is possible to use the lapping method with fixed abrasive, having significant advantages, on diamond lapp ASN10/7–B3-01–2.Studies have shown that during lapping of LNSC plates, a deformed layer of material is formed on the surface layer as a result of the impact of abrasive grains. This layer must be removed during the subsequent polishing operation. The deformed layer depthFis determined by the ratio: F = 12·h, wherehis the relief layer height. The relief layer height is characterized by the Rmax of the processed surface roughness. Rmax can be rapidly measured by a portable profiler in a production environment.The effectiveness of polishing process using Microlab Reset and VelTex Buehler non-woven polishers and SK-8 artificial leather has been proven. The diamond polishing suspension granularity lower limit is determined to be 1/0.5 µm. The processing modes are recommended: polisher rotation speed is 180–220 rpm and the pressure is 80–100 kPa, which ensures the required quality and high processing productivity.The formula recommended calculating the total depth of the deformed layer by lapping and the minimum polishing time required for its removal: F = 12·Rmax, µm; Tpol ≥ F/Qpol., min.


## Figures and Tables

**Figure 1 materials-14-04968-f001:**
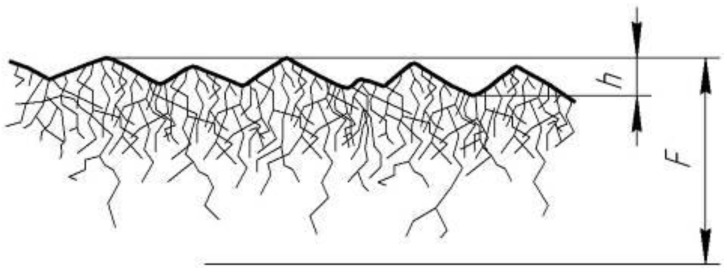
Structure of the deformed layer during lapping of brittle materials.

**Figure 2 materials-14-04968-f002:**
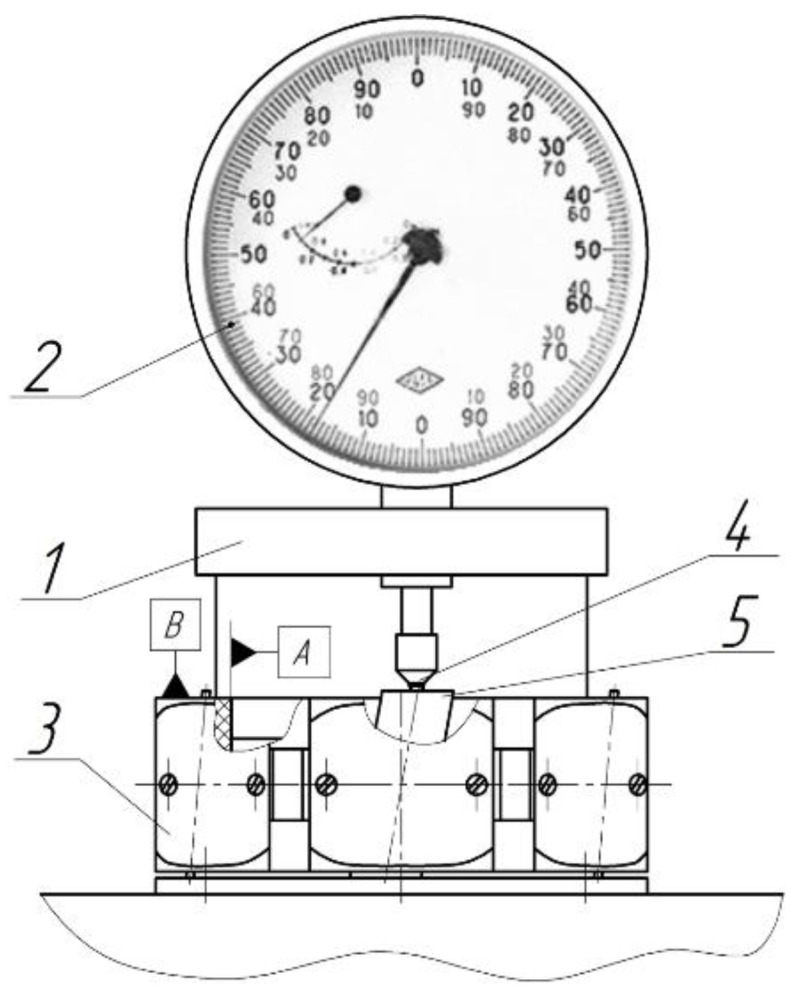
Device for measuring the amount of material removal: 1—case; 2—indicator head; 3—fixture; 4—tip; 5—package of plates.

**Figure 3 materials-14-04968-f003:**
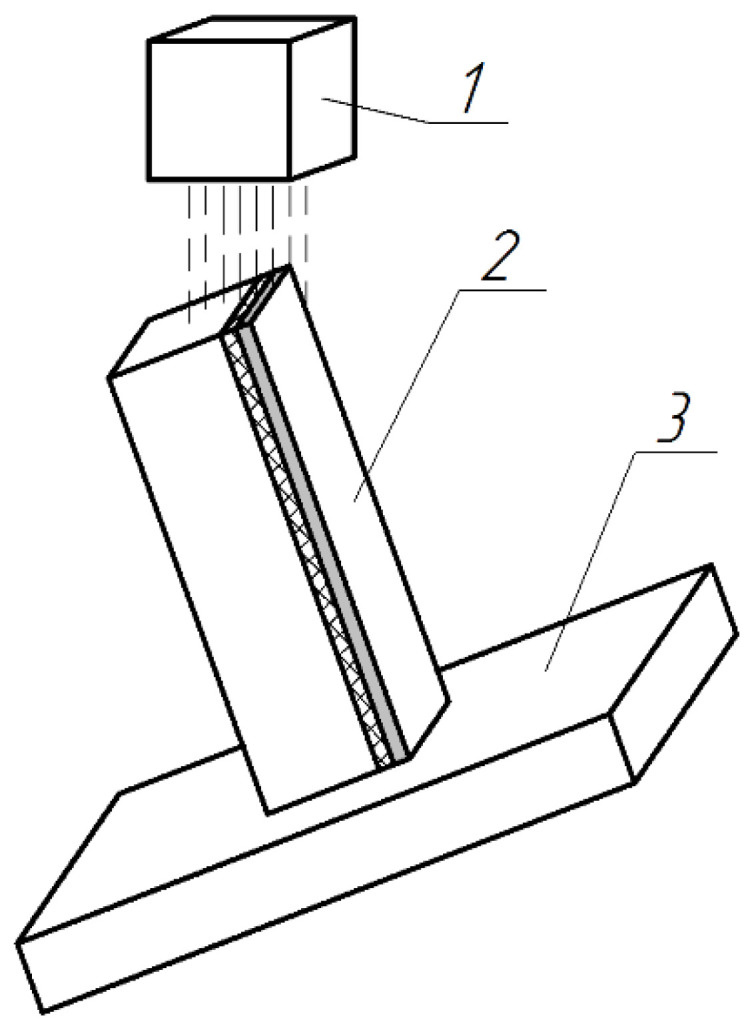
Installation of the sample during scanning: 1—detector; 2—sample; 3—the table.

**Figure 4 materials-14-04968-f004:**
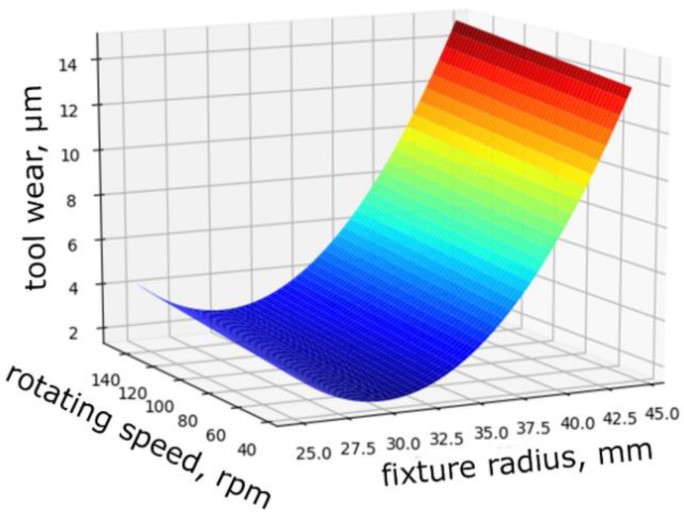
Response surface graph interaction between fixture radius and rotating speed.

**Figure 5 materials-14-04968-f005:**
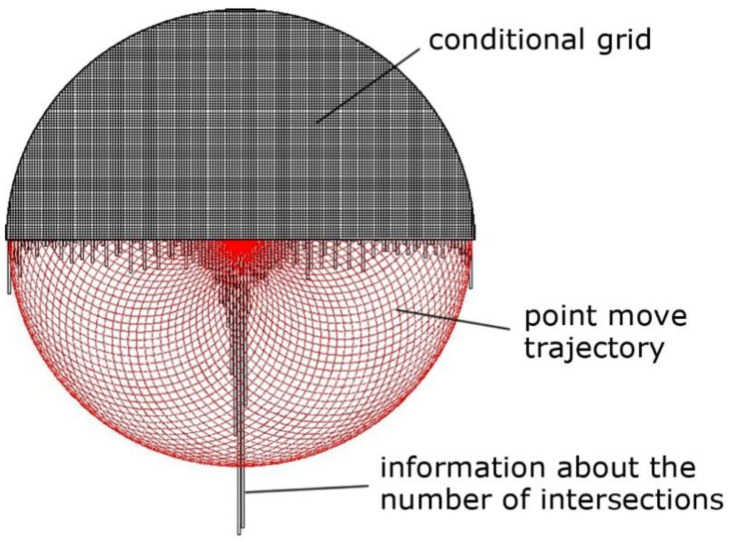
Accumulation of information about the number of intersections in conditional grid cells.

**Figure 6 materials-14-04968-f006:**
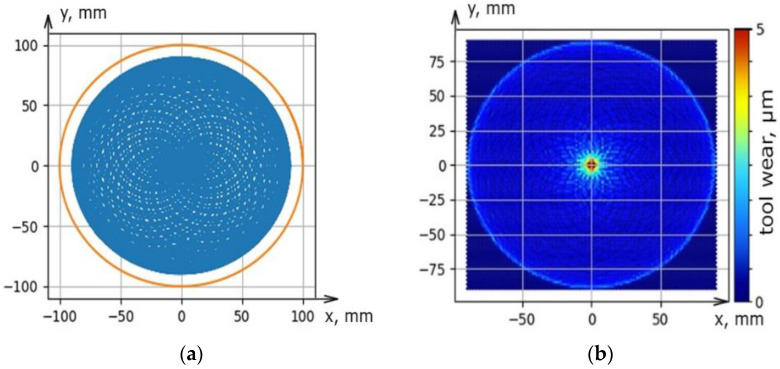
The result of modeling the trajectory distribution (**a**) and tool wear (**b**) on flat and tool wear in the cross-section (**c**).

**Figure 7 materials-14-04968-f007:**
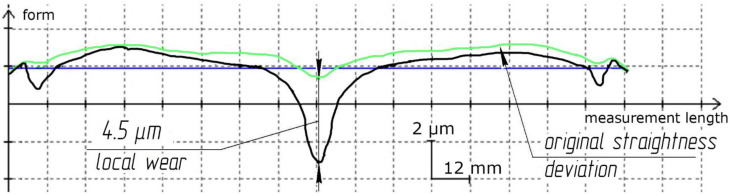
The lapp wear value as a result of real processing conditions.

**Figure 8 materials-14-04968-f008:**
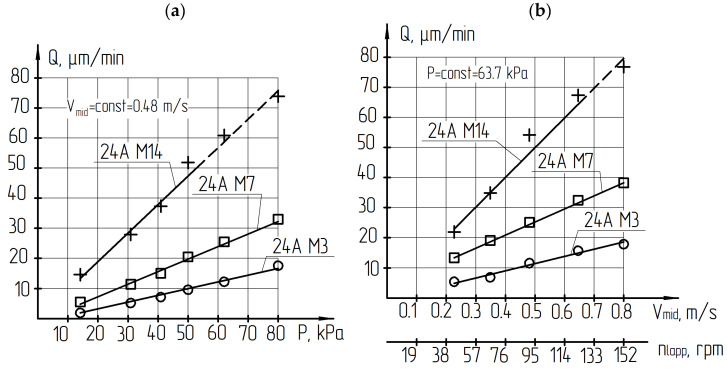
Influence of pressure (**a**) and mid (mean) cutting speed (**b**) on the lapping with a free abrasive productivity.

**Figure 9 materials-14-04968-f009:**
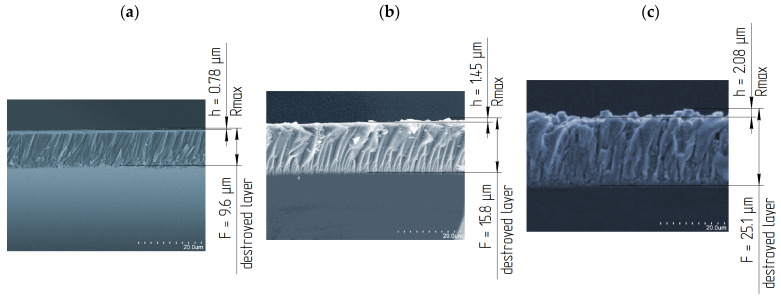
Deformed layer of LNSCplates after lapping on glass lapp of the LK5 mark with 24A suspensions, grain size: (**a**) M3; (**b**) M7; (**c**) M14.

**Figure 10 materials-14-04968-f010:**
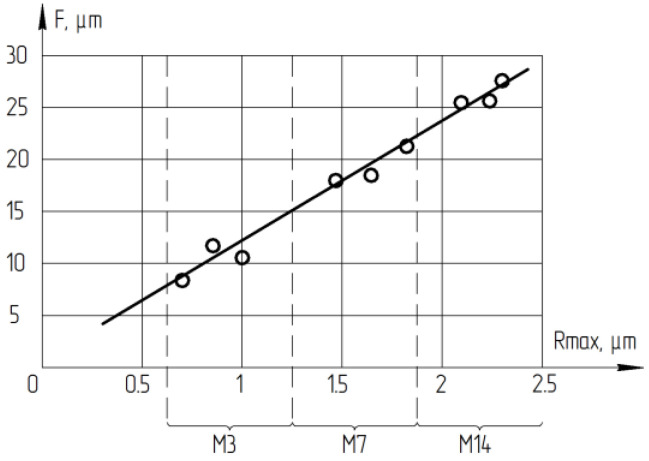
Dependence of the deformed layer depth on the height of the relief layer *h* (Rmax).

**Figure 11 materials-14-04968-f011:**
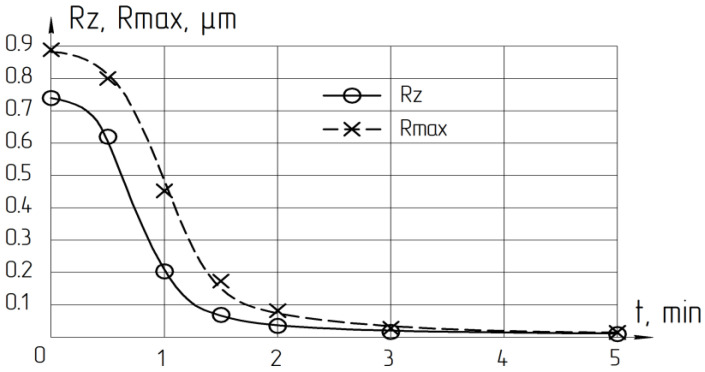
Polishing cycle time influence on the surface roughness of LNSC.

**Table 1 materials-14-04968-t001:** Kinematics influence on the parameters of the lapping process.

Trajectory		Rotation 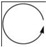	Cycloid 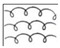	Rastr 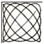	Roughness and Productivity
	Tool
Free abrasive24A M7	0.173	0.183	0.170	Ra, µm
1.322	1.375	1.215	Rz, µm
1.491	1.556	1.555	Rmax, µm
7.1	9.4	10.4	*Q*, µm/min
Fixed abrasiveASN 10/7	0.078	0.074	0.088	Ra, µm
0.674	0.553	0.694	Rz, µm
0.852	0.669	0.809	Rmax, µm
8.4	11.4	12	*Q*, µm/min

**Table 2 materials-14-04968-t002:** Influence of abrasive granularity on the lapping process parameters.

Abrasive Granularity	Ra, µm	*Q*, µm/min	Technological and Kinematic Lapping Parameters
24A M3—1/5	0.079	5.09	Lapp K8, P = 63,7 kPa, n_lapp_ = 96 rpm
24A M7—1/5	0.186	10.70	
24A M14—1/5	0.334	23.40	

**Table 3 materials-14-04968-t003:** Results of lapping of LNSC plates with a free and fixed abrasive.

Parameter of the Abrasive Layer	Roughtness, µm			*Q*, µm/min
	Ra	Rz	Rmax	
Suspension 24A M3—1/5	0.08	0.60	0.70	5.1
Suspension 24A M7—1/5	0.19	1.43	1.90	11.3
Lapp ASN 10/7—B3-01—2	0.07	0.52	0.61	10.2
Lapp ASN 10/7—M2-01—2	0.13	1.02	1.28	7.8

**Table 4 materials-14-04968-t004:** Polisher material influence on the indicators of the polishing process.

Polisher Material		Roughness, µm			*Q*, µm/min	Durability Σ*t*_mach_, min
		Ra	Rz	Rmax		
Woven	Raincoat fabric (Bologna)	0.004	0.014	0.017	3,23	~40–50
	Raincoat fabric (80% polyamide, 20% polyurethane)	0.002	0.011	0.013	2.97	
	Awning synthetic fabric with a water-repellent coating	0.002	0.013	0.014	3.17	
	Nylon mesh with a thickness of 0.09 mm, cell size ~0.1 mm	0.001	0.010	0.012	0.61	~6–10
Non-woven	Polyethylene wrap with a thickness of 0.14 mm	0.001	0.009	0.011	1.78	~6–10
	Synthetic leather SK-8	0.001	0.005	0.007	1.12	more 120
	MicrolapRemet and VelTex Buehler polisher	0.001	0.006	0.006	2.91	

## Data Availability

The authors have all the data and materials of the study.

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
