# Peer review of "Study of Lapping and Polishing Performance on Lithium Niobate Single Crystals"

_materials, 2021, doi:10.3390/ma14174968_

Round 1

Reviewer 1 Report

The paper entitled "Lithium niobate single crystals finishing precision lapping and polishing process studies" provide useful information.

The paper is Ok but it has an unusual problem:

1. Defining problem statement. 

2.The author mentioned scientists without any article references

3.In the result section, the authors presented their results but no reference to support their results.

So, minor revision to resolve above ref. issues

Author Response

Authors sincerely thank the reviewer(s) for their valuable comments and suggestions that helped to improve the quality of the paper.

The article entitled "Single crystals of lithium niobate studies of the process of precision lapping and polishing" contains useful information. The manuscript is fine, but with an unusual problem:

Observation 1

  1. Definition of the problem statement.

Response: The necessary changes have been made

Observation 2

  1. The author mentioned scientists without references to articles.

Response: The references are added supporting scientists finding.

Observation 3

  1. In the results section, the authors presented their results, but without a link to their results.

Response: The supporting references are added and linked the results

Reviewer 2 Report

In recent years, the research of integrated photonics based on Lithium Niobate Thin film(LNTF) devices is developing in full swing. A key procedure in the production of LNTF is the lapping and polishing of LNSC plates.In this manuscript, the author discusses the lapping and polishing process model of LNSC in detail, which shows the possibility of achieving the required surface quality parameters of the LNSC plates based on the mechanization of polishing. So, this manuscript is innovative and practical, which is suitable for publication on Materials.

However, there are a few writing problems in the manuscript that need to be corrected. The details are as follows:

There is a writing error in the text, for example, “1 mkm” in 129 lines, “relief layers” in 338 lines, etc.

The decimal point representation in the chart is inconsistent with that in the text. Please unify the representation.

According to the data in the last line of Table 4, non-woven is used for polishing, and its shortest polishing time t > F / Q = 12 * Rmax / Qpol = (12 * 0.006 um) / (2,91 um / min) = 0.025 min, which is incredible. Is the unit of Qpol wrong?

The format of references is not unified, for example, the abbreviation of authors’ name, the order of volume number and year, etc.

Author Response

Authors sincerely thank the reviewer(s) for their valuable comments and suggestions that helped to improve the quality of the paper.

In recent years, research on integrated photonics based on thin-film lithium niobate (LNTF) devices has been developing at full speed. The key procedure in the production of LNTF is the lapping and polishing of LNSC plates. In this manuscript, the author discusses in detail the model of the LNSC lapping and polishing process, which shows the possibility of achieving the required parameters of the surface quality of LNSC plates. based on the mechanization of polishing. So, this manuscript is innovative and practical, which is suitable for publication in Materials.

However, there are several spelling problems in the manuscript that need to be corrected. The details are as follows:

Observation 1

  1. There is a spelling error in the text, for example, "1 microns" in 129 lines, "relief layers" in 338 lines, etc.

Response: The manuscript is checked and necessary changes have been made.

Observation 2

The representation of the decimal point on the diagram does not correspond to that in the text. Please unify the presentation.

Response: The representation of decimal point is consistent throughout the manuscript

Observation 3

According to the data of the last row of Table 4, non-woven material is used for polishing, and its shortest polishing time is t> F / Q = 12 * Rmax / Qpol = (12 * 0.006 microns) / (2.91 microns / min) = 0.025 min, which is incredible. Incorrect Qpol unit?

Response:

The value of the depth of the destroyed layer F is substituted into the expression for calculating the minimum polishing time Tpol ≥ F / Qpol. The depth of the destroyed layer F is calculated by the formula: F = 12•Rmax, µm. The value of Rmax in this case should not be selected from Table 4 (Polisher material influence on the indicators of the polishing process), but from table 3 (Results of lapping LNSC with a free and fixed abrasive).

For example.

After finishing with a free abrasive, the value of Rmax=1.90 µm.

The depth of the destroyed layer in this case: F = 12•1.9 µm = 22.8, µm.

Minimum polishing time using non-woven material: Tpol ≥ 22.8 µm / 2.91 µm/min = 7.83 min.

Observation 4

The format of references is not unified, for example, the abbreviation of the authors ' name, the order of the volume number and year, etc.

Response:

The formatting of reference is according to the guidelines of Journal.
